# Evidence-Based Policy Learning

**Jann Spiess**                                                                    JSPIESS@STANFORD.EDU
*Stanford University*
**Vasilis Syrgkanis**                                                              VASY@MICROSOFT.COM
*Microsoft Research*

**Editors:** Bernhard Schölkopf, Caroline Uhler and Kun Zhang

## Extended Abstract

The past years have seen the development and deployment of machine-learning algorithms to estimate heterogeneous treatment effects and derive personalized treatment-assignment policies from randomized controlled trials. These algorithms hold the promise of identifying groups of individuals with positive treatment effects. Rather than just estimating *whether* some treatment is effective, we can use these tools to answer the question *for whom* an intervention works and *who* should get treated.

In many applications, finding individuals with positive treatment effects involves an additional hypothesis-testing step: after having identified a group of individuals, we may have to provide evidence that the treatment has a positive effect on this group, or equivalently, that the assignment produces a positive net effect relative to some status quo. For example, a drug manufacturer may have to demonstrate that the drug is effective on the target population by submitting the result of a hypothesis test to the FDA for approval. Existing algorithms for the assignment of treatment typically optimize policy values without taking hypothesis testing into account.

We consider assignments that take subsequent hypothesis testing into account and directly optimize the probability of finding a subset of individuals with a statistically significant positive treatment effect. Specifically, we consider an offline learning problem with two datasets drawn from the same distribution. Treatment is assigned randomly in both datasets. On the first dataset (training), we estimate an assignment policy that maps features to treatment. On the second dataset (hold-out), we run a hypothesis test to reject the null hypothesis that this assignment policy is no better than some reference policy (such as assigning everybody to control). We propose a set of algorithms for the first step (training) that maximizes the probability of rejecting the hypothesis test in the second step.

We show that the optimal policy for maximizing the probability of rejection (power) of the hold-out test differs from optimal policies that maximize the policy value. When target groups differ not only in their treatment effects, but also in how precisely these treatment effects can be measured in a trial, then the additional testing step may affect the optimal choice of treatment assignment. This is because the power-maximizing test trades off the policy value with the variance in its estimation on the hold-out set, leading us to prefer not to assign units to treatment that have very noisy outcomes, even if their estimated treatment effect is positive.

We provide an efficient implementation using decision trees, and demonstrate its gain over selecting subsets based on positive (estimated) treatment effects. Compared to standard tree-based regression and classification tools, this approach tends to yield substantially higher power in detecting subgroups affected by the treatment. We consider a continuous relaxation with randomized treatments, as well as extensions to observational studies, heterogeneous treatment effects, and alternative objective functions that integrate hypothesis testing and the policy value.

**Keywords:** Policy learning, heterogeneous treatment effects, randomized experiments, recursive partitioning, causal machine learning

**Full version:** arxiv.org/abs/2103.07066

