# OpenReview forum: "Evidence-Based Policy Learning"
_cclear.cc/CLeaR/2022/Conference — CLeaR 2022 Oral_

### Official Review · Reviewer_EQsk · 2021-11-23

**Confidence:** 3
**Overall Score:** 7

**Main Review:**

High level overview of comments:
-	I would suggest a more through discussion of relevant literature, especially the literature on policy improvement (e.g. https://proceedings.mlr.press/v37/thomas15.html) and pure exploration (e.g. https://arxiv.org/abs/1507.04523), which are also concerned with ensuring estimation guarantees. Discussing how your work relates to this literature would help the reader better put your work in context.
-	Additionally, the writing can be clearer early on (i) what problem you are approaching (e.g., online vs offline setting, defining training set) and (ii) why an optimal policy for your problem is different from the optimal policy for minimizing regret.

More detailed comments:

-	Regarding the abstract: “Yet such algorithms typically optimize expected outcomes without taking into account that treatment assignments are frequently subject to hypothesis testing.” What does hypothesis testing of the treatment assignments mean? I thought in a randomized control trial with binary treatments the treatment assignments are most commonly selected with probability 0.5 each---why are there a hypothesis tests on these treatment assignments?
-	Whey you say “we assume that a policy obtained from training data only gets implemented when a hypothesis test…” what do you mean by implemented here? How is implementation related to the training data?
-	From reading your introduction I am confused about whether you are (i) using batch training data to estimate what policy you want to use to collect more data in order to maximize the probability of having a significant hypothesis test or (ii) you are in an online setting in which you are changing treatment assignment probabilities (in reference to the sentence “existing algorithms for the assignment of treatment typically optimize welfare without taking hypothesis testing into account”, which makes me think you in an online setting). It would be helpful for the reader to better understand at a high-level what problem setting you are in.
-	In your related work section, you do not mention literature on best arm identification and algorithms for pure exploration, which seem highly relevant. For example, see https://homes.cs.washington.edu/~jamieson/resources/bestArmSurvey.pdf, https://arxiv.org/abs/1507.04523, https://arxiv.org/abs/0802.2655, https://arxiv.org/pdf/2101.08534.pdf, https://maxkasy.github.io/home/files/papers/adaptiveexperimentspolicy.pdf . Is your method a non-online version of best arm identification? This literature is not concerned just with “optimizing welfare”, but very much take hypothesis testing into account.
-	In your introduction you say that you compare your algorithm to “algorithms that target maximal average outcome by assigning based on positive estimated treatment effects.” I wonder if there are better baselines to compare to because this is a bit like comparing a best arm identification algorithm to a regret minimizing bandit algorithm on the task of best arm identification. Would also comparing to Dwivedi et al. (2020) and/or Leqi and Kennedy (2021) make sense? You say that these works don’t directly target out-of-sample power as an objective but their methods could “lead to better out-of-sample power of statistical tests”, which makes me think they would be reasonable to compare to.
-	As I read into section 2, I want to confirm my understanding of the problem setup. I understand that you have n i.i.d. observations to train on and a hold-out set that you will eventually use to perform a hypothesis test. You are trying to estimate some optimal policy a^* using estimated policy \hat{a}, which is most likely to be significant if we compare treatment \hat{a} to the baseline treatment according to the hypothesis test on the held-out data. Based on my understanding, this is different from the problem of estimating an optimal policy regret minimizing policy from batch data, because the optimal policy for your problem will be a function of the noise variance in each state (which relates to the power of a test), while this does not matter for the problem of estimating optimal regret minimizing policies. If this is the case, it would be helpful to provide this intuition to the reader early on to prevent confusion.
-	The presentation of Algorithm2 is hard to read. It is not in the typical format of an algorithm, but instead one long paragraph.


**Summary:**

This paper works on the problem of using batch data to estimate a policy that on average has a high probability of being significantly better than a baseline policy. They provide asymptotic and finite sample guarantees for their approach.

---

> ### Author Response · Authors · 2021-12-04
> **Response to EQsk**
>
> Thank you for engaging with our draft and for your extensive comments, suggestions, and questions.
>
> > Additionally, the writing can be clearer early on (i) what problem you are approaching (e.g., online vs offline setting, defining training set) and (ii) why an optimal policy for your problem is different from the optimal policy for minimizing regret.
>
> We appreciate the suggestion for additional clarity. Would the below formulation in response to your two high-level points be helpful?
>
> (i) We approach an offline learning problem with two datasets drawn from the same distribution. Treatment is assigned randomly in both datasets.
> On the first dataset (training), we estimate an assignment policy that maps features to treatment.
> On the second dataset (hold-out), we run a hypothesis test to reject the null hypothesis that this assignment policy is no better than some reference policy (such as assigning everybody to control).
> We propose a set of algorithms for the first step (training) that maximizes the probability of rejecting the hypothesis test in the second step.
> This procedure is motivated by a desire to find an assignment policy with significant evidence for its efficacy, for example when regulatory constraints require the rejection of a hypothesis test.
>
> (ii) The optimal policy for maximizing the probability of rejection (power) of the hold-out test differs from optimal policies that maximize the policy value or minimize regret. This is because the power-maximizing test trades off the policy value with the variance in its estimation on the hold-out set, leading us to prefer not to assign units to treatment that have very noisy outcomes, even if their estimated treatment effect is positive. While some other objectives may also involve estimation noise, we are not aware of other procedures that trade off policy value and variance in the same way.
>
> > [...] What does hypothesis testing of the treatment assignments mean? I thought in a randomized control trial with binary treatments the treatment assignments are most commonly selected with probability 0.5 each---why are there a hypothesis tests on these treatment assignments?
>
> Thank you for the clarification question. What we mean here (and should have written) is that we frequently test whether the value (expected outcome) achieved by an assignment policy is above some outside option (such as assigning everybody to treatment). So we are not evaluating the random assignment in the data. Instead, we estimate the policy value on the hold-out data and run a hypothesis test there.
>
> > [...] what do you mean by implemented here? How is implementation related to the training data?
>
> The assignment policy is estimated on the training data. If it passes a hypothesis test on the hold-out data, we imagine that the policy is finally implemented on new data, that is, new units from the population distribution are assigned according to the policy in that case.
>
> > From reading your introduction I am confused about whether you are (i) using batch training data [...] or (ii) you are in an online setting [...]
>
> Thanks for this suggestion. We are talking about an offline setting where training and hold-out data are given, both are drawn from the same distribution, and treatment is assigned randomly in both. This models a case of existing data from a randomized controlled trial (e.g. in medicine), which is our motivation.
>
> > In your related work section, you do not mention literature on best arm identification and algorithms for pure exploration, which seem highly relevant. [...] Is your method a non-online version of best arm identification? [...]
>
> Thank you for suggesting additional references. We differ from best-arm identification not only in that we consider a non-online setting, but also in the specific objective we use, which is maximizing the power of a frequentist hypothesis test on the hold-out. We agree that alternative (especially online) approaches also take uncertainty into account, although to our knowledge they do not directly target the probability of rejection of a hold-out test.
>
> > [...] Would also comparing to Dwivedi et al. (2020) and/or Leqi and Kennedy (2021) make sense? [...]
>
> We agree, thank you for this suggestion. We chose what we perceived to be the most common methods for treatment assignment as comparisons, and agree that the methods you suggest may be more relevant for the objective at hand.
>
> > As I read into section 2, I want to confirm my understanding of the problem setup. [...] If this is the case, it would be helpful to provide this intuition to the reader early on to prevent confusion.
>
> We agree with your understanding, and appreciate the suggestion.
>
> > The presentation of Algorithm2 is hard to read. It is not in the typical format of an algorithm, but instead one long paragraph.
>
> Thank you for catching this. Both algorithms are missing line breaks that should be there, which seems to be an issue with LaTeX.

---

> > ### Comment · Reviewer_EQsk · 2021-12-19
> > **Re: Rebuttal**
> >
> > Thank you for your responses and clarifications!
> >
> > "Would the below formulation in response to your two high-level points be helpful?" Yes, I like the explanations you wrote above.
> >
> > I have raised my score accordingly.

---

### Official Review · Reviewer_PGmN · 2021-11-24

**Confidence:** 3
**Overall Score:** 8

**Main Review:**

The originality seems to be in using the tests as split criteria. I am not familiar with the literature on competing methods but this seems well done.

**Summary:**

The paper proposes a heuristic method for finding subgroups of a population for which a treatment is most beneficial, using a decision tree with splits determined by statistical tests (e.g., t tests) applied to a hold-out sample.

---

> ### Author Response · Authors · 2021-12-04
> **Response to PGmN**
>
> Thank you for your comment.
>
> We agree that the main innovation in our approach is that model selection itself (here, the splits of the tree) as well as the assignment given the model are driven by the objective of rejecting a hypothesis test on the hold-out data.

---

### Official Review · Reviewer_KuP2 · 2021-11-24

**Confidence:** 3
**Overall Score:** 6

**Main Review:**

The paper studies the hypothesis testing problem in learning
treatment-assignment policies. Specifically, they argue that we
should optimize for an assignment with maximal expected probability of
passing the test. The goal is to help a drug company to get a drug
approved, or help a researcher to find significantly positive
treatment effect. The post-selection issue is handled by sample
splitting.

This goal, though seemingly a little eye-brow raising at first, aligns
with active learning or Bayesian optimization, where we want to find
the data points that are most helpful for reducing the desired
prediction or optimization loss.

The optimization objective is straightforward, though optimizing for
it is challenging. In particular, the target of inference is a
function, which could be hard to handle if it relies on many or
high-dimensional covariates. Thanks to this concern, the paper
developed a solution based on submodular optimization, which is
interesting, though I am not an expert in submodular optimization to
judge its soundness.


The derivation of the paper seems sound. However, one thing that could
be better clarified is its connection to Bayesian optimization.
Bayesian optimization seems to be solving a related and possibly
harder problem. Striping away the sequential acquisition nature of
Bayesian optimization, I can think of Bayesian optimization as finding
the set of data points such that the resulting target (which is the
probability of passing the test here) is maximized. I can think of
finding treatment policy as a form of selection data points (e.g. we
have pool of data points with the counterfactual outcomes of all
possible treatment configurations.) In this framing, does the proposed
algorithm outperform standard Bayesian optimization algorithm (using
only one acquisition step of Bayesian optimization)? How is the
proposed algorithm better? I think the paper could benefit from
clarifying this point.

**Summary:**

review

---

> ### Author Response · Authors · 2021-12-04
> **Response to KuP2**
>
> Thank you for your comments and your suggestion to relate our work to Bayesian optimization. We agree that it may be interesting to consider Bayesian approaches for the target we motivate and optimize for, but believe that a more thorough treatment of Bayesian optimization would be beyond the scope of the current draft.
>
> One specific Bayesian approach that targets maximizing the power of the (frequentist) test in the hold-out could take the following form: From the training data, form a posterior about the population distribution. Then obtain an optimal assignment given the features in the test data by maximizing the posterior expectation over our objective (that is, rejecting the test).
> We note that such a procedure would have to form a posterior about the distribution of outcomes conditional on features that goes beyond conditional means, since our objective takes a non-linear form (and depends, in particular, on the conditional variances). We further would then face the challenge of optimizing the posterior expectation over all binary assignments. So while we agree that Bayesian optimization could be an alternative to the regularized greedy optimization we currently perform to find an assignment, we believe that such an implementation is non-trivial.
>
> As an alternative interpretation, it could also be relevant to discuss the performance of Bayesian treatment assignment strategies, even if they do not directly target the power of a (frequentist) hypothesis test on the hold-out data.
>
> Please let us know in case you were thinking of a simpler procedure, or if we misunderstood your suggestion.

---

### Decision · Program_Chairs · 2022-01-13

**Decision:**

Accept (Oral)

**Comment:**

Reviewers are largely positive about this paper. The author response has successfully removed some of the doubts and answered some of the questions. We encourage the authors to use the reviewer feedback to further improve the presentation and clarity, especially considering that their approach is somewhat non-standard.